# Management and Rehabilitative Treatment in Osteoarthritis with a Novel Physical Therapy Approach: A Randomized Control Study

**DOI:** 10.3390/diagnostics14111200

**Published:** 2024-06-06

**Authors:** Teresa Paolucci, Marco Tommasi, Giannantonio Pozzato, Alessandro Pozzato, Letizia Pezzi, Mariachiara Zuccarini, Alessio Di Lanzo, Rocco Palumbo, Daniele Porto, Riccardo Messeri, Mirko Pesce, Andrea Pantalone, Roberto Buda, Antonia Patruno

**Affiliations:** 1Department of Oral, Medical and Biotechnological Sciences, Physical Medicine and Rehabilitation, G. D’Annunzio University of Chieti-Pescara, 66100 Chieti, Italy; teresa.paolucci@unich.it (T.P.); mariachiara.zuccarini@unich.it (M.Z.); alessio.dilanzo@gmail.com (A.D.L.); 2Department of Medicine and Aging Sciences, G. D’Annunzio University of Chieti-Pescara, 66100 Chieti, Italy; marco.tommasi@unich.it (M.T.); andrea.pantalone@unich.it (A.P.); roberto.buda@unich.it (R.B.); antonia.patruno@unich.it (A.P.); 3Telea Electronic Engineering srl, 36066 Sandrigo, Italy; gianantonio.pozzato@teleamedical.com (G.P.); alessandro.pozzato@teleamedical.com (A.P.); 4Unit of Physical Medicine and Rehabilitation, Hospital of Cremona, 26100 Cremona, Italy; pezziletizia@gmail.com; 5Department of Psychological, Health and Territorial Sciences, G. D’Annunzio University of Chieti-Pescara, 66100 Chieti, Italy; rocco.palumbo@unich.it; 6Institute Don Orione, Medical-Social Recovery Center, 65128 Pescara, Italy; daniele.porto@virgilio.it (D.P.); messeri.riccardo84@gmail.com (R.M.); 7UdA-TechLab, Research Center, University of Chieti-Pescara, 65100 Pescara, Italy

**Keywords:** inflammation, physiotherapy, rehabilitation, physical therapy, arthrosis, pain

## Abstract

Knee osteoarthritis (KOA) is a chronic degenerative disease characterized by progressive joint damage leading to significant disability. Although rehabilitative treatment methods for KOA have been widely implemented, the optimal integrated instrumental physical therapy approach remains unclear. Therefore, this study aimed to analyze the effect of Quantum Molecular Resonance (QMR) on pain reduction as the primary outcome and the functional improvement in activity daily living (ADL) as a secondary outcome. The study was designed as a double-blind, randomized, controlled trial in an outpatient setting. Fifty-four (*N* = 54) patients were enrolled and then randomized into three groups according to a simple randomization list: Group 1 (intensive protocol, *N* = 22), Group 2 (extensive protocol, *N* = 21), and a Sham group (*N* = 11). Patients were evaluated over time with scales assessing pain and function. Treatment was performed with the QMR model electro-medical device, which generates alternating electric currents characterized by high frequency (4–64 MHz). The results showed that QMR had a positive effect with respect to the Sham group in terms of pain and function (*p* < 0.01), and intensive treatment was more effective than the extensive treatment in terms of “speed of response” to the treatment (*p* < 0.05). In conclusion, QMR in KOA could be effective in slowing the progression of clinical symptoms and improving patients’ pain and functionality and thus quality of life. Future studies will be necessary to investigate further treatment algorithms and therapeutic associations with rehabilitative exercise.

## 1. Introduction

Knee osteoarthritis (KOA) is a chronic degenerative disease, characterized by progressive joint damage until the appearance of significant disability, and although rehabilitative treatment methods for KOA have been widely implemented, the optimal integrated instrumental physical therapy approach remains unclear [1,2,3,4].

The World Health Organization (WHO) classifies this pathology as the eleventh cause of disability. According to the Osteoarthritis Research Society International (OARSI), at least 40% of people over age 65 have symptomatic osteoarthritis of the hip or knee. KOA is a pathology typical of advanced age (60 years), especially for its primary forms (i.e., of unknown cause), which are more prevalent in the female sex. When arthrosis arises from a pre-existing morbid condition (e.g., obesity, predisposing factors, sequelae of knee joint fractures, systemic causes), defined as secondary, the average age of onset drops considerably (40–50 years). Pathological changes typical of KOA patients, including articular cartilage degradation, synovial inflammation, and subchondral bone thickening, are responsible for pain and disability in KOA patients [1,5,6].

Articular cartilage alteration is the most common symptom of patients with KOA. In pathological conditions of KOA, an activation of quiescent chondrocytes is observed, characterized by increased cell proliferation and secretion of inflammation-related cytokines and matrix-degrading enzymes [2]. The main objectives of the treatment of KOA are aimed at improving/maintaining mobility and function, the relief of pain and inflammation, and the prevention of decline in quality of life. The guidelines for KOA are the broadest and most numerous found in the literature, indicative of the fact that knee osteoarthritis is a frequent and disabling disease with increased incidence and prevalence in the general population.

The European League against Rheumatism (EULAR) and the American College of Radiology (ACR), specify treatment guidelines, respectively comprising pharmacological and rehabilitation treatments in force from 2003 and non-pharmacological rehabilitation treatments established in 2023 [7,8,9,10]: the recommendations include a personalized and multicomponent management plan including information, education, self-management, and exercise with adequate tailoring of dosage and progression; mode of exercise delivery; maintenance of healthy weight and weight loss; footwear, walking aids, and assistive devices; and work-related advice and changes to sedentary and unhealthy lifestyles. Furthermore, it is important to specifically consider the patient’s needs in proposing manageable protocols, such as commitment and timing as well as work activity, in order to improve adherence to rehabilitation treatment [10,11].

Although treatment methods for KOA have been widely implemented and studied, the optimal treatment or treatment combinations remain unclear. Therefore, innovative and safe strategies aimed at reducing the inflammatory process and promoting tissue regeneration events are necessary to inhibit the progression of KOA.

Inflammation is an important factor in the pathogenesis and progression of KOA. Although often subacute, KOA-related inflammation is recognized by methods such as magnetic resonance imaging (MRI) and ultrasound. Innate immune cells, including macrophages, are important in driving inflammation and destructive responses in KOA [8,10]. In KOA synovial tissues, the predominant presence of both macrophage cells and T-helper cells (Th1 and Th2) has been demonstrated. Macrophages are immune cells that play a crucial role in innate immunity and participate in tissue repair and remodeling [12].

Macrophages can be polarized into two functionally different phenotypes (M1 and M2) in response to their microenvironment. Classically activated macrophages (M1 macrophages) are known to release inflammatory cytokines (IL-1β and TNF-α) that promote tissue damage and inflammation [13]. Alternatively activated macrophages (M2 macrophages) are known to release anti-inflammatory cytokines (IL-10) and growth factors such as TGF-β, which promote tissue remodeling and suppress inflammation [14].

Recent studies have shown that M1 macrophages are present in the synovium and synovial fluid of patients with OA and are involved in the progression of the disease, suggesting that these cells could be a target for possible treatments of the disease [15].

In in vitro experiments carried out in our laboratory, we found that Quantum Molecular Resonance (QMR) leads to the reduction of proinflammatory mediators and nitrosative stress by inhibiting the expression of COX-2 and iNOS proteins and reducing NF-κB activity and peroxynitrite levels [16].

From the above, in this study we hypothesize that the treatment with QMR found in vitro [16] could, in vivo, translate into a reduction in pain in patients with symptomatic KOA due to its anti-inflammatory and regenerative action; there are no studies on this subject in the literature.

From these premise, the main objectives of the study are (a) the evaluation of the potential therapeutic role of QMR treatment with respect to the reduction of pain as the primary outcome, (b) the recovery of function as a secondary goal, and (c) the comparison of two novel physiotherapy programs in KOA patients.

## 2. Materials and Methods

### 2.1. Study Design and Ethical Approval

This study was designed as a double-blind, randomized, controlled trial and took place at the University Gabriele D’Annunzio of Chieti, Orthopedics clinic (Italy)—14 February to 30 November 2023—to evaluate the efficacy of Quantum Molecular Resonance (QMR) in the treatment of KOA, following the Consort Guideline [17].

Fifty-seven patients (*N* = 57) who were diagnosed with KOA were referred to a physiatrist: fifty-four (*N* = 54) patients were enrolled because they met the inclusion criteria and agreed to participate, and they then were randomized into three groups: Group 1—INTENSIVE (*N* = 22), Group 2—EXTENSIVE (*N* = 21) and the Sham group (*N* = 11) (Figure 1—Flow Chart).

For concealment of the allocation, a physiatrist identified the patients who were sent by orthopedic allocation and who met the inclusion and exclusion criteria and obtained signed informed consent forms for study participation. The patients, the physiatrist who enrolled them, and the researchers (including the physiotherapist) who administered the evaluation scales were blinded to the rehabilitative treatment.

All procedures involving human participants followed the ethical standards of the institutional committee and the principles of the Declaration of Helsinki. The Local Ethical Committee approved the study (Protocol ID is 22016). The Clinical Trial Registration ID is NCT06239805.

### 2.2. Patients

Fifty-four patients of both sexes were enrolled through direct contact at the Orthopedic Clinic of the SS Annunziata Hospital in Chieti (Italy); the physiotherapy treatment was carried out at the Don Orione Institute in Pescara (Italy) by two physiotherapists with expertise in instrumental physical therapy and blinded to the treatment of the patients.

The eligibility assessment was carried out at these sites, where the medical history and clinical examinations were considered according to inclusion and exclusion criteria. The recruitment period for eligible subjects lasted up to 6 months.

Informed consent was obtained for patients considered eligible, who were then included in the randomization lists for allocation to the sham group (=Sham) or to the Experimental Treatment Group 1 or Group 2.

Both male and female subjects, aged between 40 and 88 years, with acute and chronic phase gonarthrosis, diagnosed by standard radiography, were included. In addition, diagnosis of gonarthrosis included constant pain for VAS > 3 at rest and radiographic evidence of KOA for the Kellegren–Lawrence staging score (I, II, III). Patients with the following symptoms were excluded: favism, hemolytic anemia, severe hyperthyroidism, Grave’s disease, thrombocytopenia < 50,000 and severe coagulopathy, severe cardiovascular instability, coagulation disorders, alcohol abuse; hemochromatosis, patients treated with dietary supplements, pregnancy and lactation, psychiatric disorders, less than three months after previous knee infiltration, septic arthritis and/or febrile conditions, and history of contraindications to current instrumental physiotherapy (previous cancer). In addition, subjects with rheumatic and autoimmune diseases and a recent history of trauma and/or distortions of the knee (ligaments) were excluded from the study.

The pharmacological therapeutic regimen was required to be stable for at least 3 months before the patient began treatment. No new medications or other rehabilitation approaches were undertaken during this study.

### 2.3. Treatment Rehabilitative Protocol

The protocol for the open-label study with the Sham group consisted of three study phases, with a total duration of 2 months. Patients were evaluated at T0 = before treatment; T1 = at the end of treatment; and T2 = one month after the end of treatment (follow-up).

Treatment was performed with the Q-Physio model electro-medical device (code 4001006), serial number D06164121, manufactured by Telea Electronic Engineering Srl (Italy), in the premises of the Diagnostic and Treatment Department of the Don Orione Institute in Pescara, on an outpatient basis. Patients were treated as follows: the Sham group underwent three sessions per week, for a total of 6 sessions, each lasting 30 min. The sessions consisted of the application of electrodes (adhesive plates) with the device switched off. Treatment group 1 (intensive protocol) underwent three sessions per week, for a total of 6 sessions, each lasting 30 min. The sessions included the application of electrodes (adhesive plates) with the device switched on. Treatment group 2 (extensive protocol) underwent two sessions per week, for a total of 8 sessions, each lasting 20 min. The sessions included the application of electrodes (adhesive plates) with the device active. Treatments were performed with the patient in the supine position. The floating electrode was placed between the couch and the patient’s gluteal region to maximize the contact area (Figure 2).

The device generates alternating electric currents characterized by high-frequency (4–64 MHz) and low-intensity waves. No significant temperature changes were observed to be associated with the application of the QMR generator (∆T = 0.1 °C).

Patients were asked to use a diary to record the ongoing analgesic/anti-inflammatory therapy at T0 and the end of treatment (T1).

### 2.4. Sample Size Calculations

We aimed for a balanced sample size across treatment levels (sham, extensive, and intensive) while considering outpatient pain levels. Considering a 10% dropout rate [18], with 54 valid outpatients, potentially 48.6 would remain after dropouts, split equally into three groups (16.2 per group). We opted for a 2:1 ratio between treated and control groups. Research suggests lower sample sizes do not affect statistical power [19]. Using Sakpal’s formula [20], we estimated needing 24 participants per treatment group and 12 as controls. Yet, with only 54 participants, adjustments were necessary. Thus, treatment groups could consist of 21–22 participants, and the control group of 10–11.

### 2.5. Outcome Measures

The following scales were used to assess pain and function at T0, T1, and T2: Visual Analogue Scale (VAS) [21]; Knee Injury and Osteoarthritis Outcome Scale (KOOS) [22]; Lysholm knee scoring scale [23].

#### 2.5.1. Pain Evaluation

The VAS scale is the most widely used tool for pain assessment [21] and consists of a one-dimensional rating of pain intensity; it is a continuous scale consisting of a horizontal or vertical line, generally 10 cm (100 mm) long, with two start and end points marked ‘no pain’ and ‘worst pain ever’.

#### 2.5.2. Functional Evaluation

Recovery of function was understood as improvement in symptoms, walking, common activities of daily living, and quality of life for the Knee injury and Osteoarthritis Outcome Score (KOOS-I) [22] and Lysholm Knee Scoring (Lysholm-S) [23].

The KOOS scale is administered as a self-completion questionnaire by the patient; it consists of 42 items in 5 subscales (Frequency and intensity of pain during functional activities; Symptoms like stiffness, swelling, presence of joint noise or locking ROM limitation; Difficulty with activities of daily living (ADL); Difficulty with recreational/sports activities; Knee-related quality of life) used to measure patients’ opinions about their knee problems in the short and long term (7 days to 10 years) [22].

The Lysholm Knee Scoring Scale15 is a patient-reported outcome measure (PROM) questionnaire designed to assess outcomes after knee surgery, particularly symptoms related to instability; it has 8 items, and the score obtained is on a scale from 0 to 100, with 100 representing the absence of symptoms and disability. Scores are categorized as excellent (95–100), good (84–94), fair (65–83), and poor (<64) [23].

Also, between T1 and T2, patients did not undertake any other rehabilitation procedures or take any medicines.

### 2.6. Statistical Analysis

We calculated descriptive statistics, including means, standard deviations, skewness, and kurtosis, for each psychological scale. For result interpretation, we adhered to well-established criteria in the literature. Acceptable skewness and kurtosis values should fall between ±2. Regarding Pearson correlation values, we followed Cohen’s criteria (1992) for interpretation (r < 0.30 = low; 0.30 ≤ r < 0.50 = moderate; r ≥ 0.50 = high). We conducted a 3 (time series) × 3 (gonarthrosis treatment) mixed ANOVA for each variable (VAS, KOOS, and Lysholm scale) to assess how the effect varied across three different time periods: T0, T1, and T2. We also examined variations among different groups: the Sham group, the group under an extensive protocol, and the group under an intensive protocol. Addition ally, we performed post hoc analysis (Tukey *t*-test) to identify potential significant differences between group means and conducted contrast analyses between means of different time periods to examine the form (longitudinal or quadratic) of the primary effect. A significant linear contrast indicates a constant increment or decrement of the primary effect over time, while a significant quadratic effect suggests a potential inversion (initial increment followed by a decrement or vice versa) of the primary effect over time. All analyses were conducted using JASP 0.17.1 software (JASP Team, 2023).

## 3. Results

### 3.1. Descriptive Statistics

In this study, 54 patients, including 18 males and 33 females, diagnosed with KOA were randomized and divided into three groups, with 22 subjects included in the treated Group 1, 21 subjects included in the treated Group 2, and 11 subjects included in the sham group. Patients’ descriptive findings are shown in Table 1, Table 2 and Table 3.

### 3.2. Inferential Statistics

#### 3.2.1. Power Analysis

We calculated with G-power the value of the effect size in relation to our sample size. According to Cohen (1992) for ANOVA designs, the three different levels of effect sizes (small, medium, and large) correspond to f values of 0.02, 0.15, and 0.35. With α = 0.05 (first type error) and 1 − β = 0.80 (power), with a sample of 51 participants and a mixed 3 × 3 ANOVA model, the resulting effect size is f = 0.202, which indicates a medium effect size (Table 4).

#### 3.2.2. Mixed-Effects ANOVA-Dependent Variable: VAS Scores

The principal effects of time series and treatment of gonarthrosis are both significant (F_2,96_ = 45.061 and F_2,48_ = 11.306, with *p* < 0.001, respectively). Mean ratings of VAS decrease over time and in extensive and intensive treatment groups. Interaction of time series × treatment of gonarthrosis is significant (F_4,96_ = 10.055, *p* < 0.001). The reduction in VAS scores over time is present in the groups with extensive or intensive treatment but not in the sham group (Table 5).

#### 3.2.3. Mixed-Effects ANOVA-Dependent Variable: KOOS-I Scores

The principal effects of time series and treatment of gonarthrosis are both significant (F_2,96_ = 32.973 and F_2,48_ = 3.953, with *p* < 0.001 and *p* = 0.026, respectively). Mean ratings of KOOS-I increase over time and in extensive and intensive treatment groups. Interaction of time series × treatment of gonarthrosis is significant (F_4,96_ = 12.462, *p* < 0.001). The increment in KOOS-I scores over time is present in the groups with extensive or intensive treatment but not in the sham group (Table 6).

#### 3.2.4. Mixed-Effects ANOVA-Dependent Variable: Lysholm-S Scores

The principal effects of time series and treatment of gonarthrosis are both significant (F_2,96_ = 46.763 and F_2,48_ = 5.557, with *p* < 0.001 and *p* = 0.007, respectively). Mean ratings of Lysholm-S increase over time and in extensive and intensive treatment groups. Interaction of time series × treatment of gonarthrosis is significant (F_4,96_ = 12.895, *p* < 0.001). The increment in Lysholm-S scores over time is present in the groups with extensive or intensive treatment but not in the sham group.

Table 7 shows that the intensive treatment is the most efficient in the post hoc analysis, because mean differences between control and intensive group marginal means are significant at T1 and T2 and have larger effect sizes (Cohen’s d) in comparison to the differences between control and extensive groups. Before-treatment (T0) post hoc analysis revealed no differences between groups.

#### 3.2.5. Trend Analysis for VAS, KOOS-I, and Lysholm-S Scores

Table 8 presents linear and quadratic contrasts for VAS, KOOS-I, and Lysholm-S scores over time. Except for KOOS-I, both linear and quadratic trends are statistically significant. However, it is worth noting that linear contrasts tend to exhibit larger effect sizes. This suggests that the treatment effect predominantly has a steady increment or decrement over time, and this is also maintained in the follow-up period (Figure 3).

## 4. Discussion

This study aimed to analyze the effect of QMR on pain reduction in patients with KOA and, as a secondary outcome, to measure the improvement of function in reference to the activities of daily living. The results obtained on both outcomes were, respectively, very encouraging for pain and function; QMR indeed proved effective after physiotherapy (at T1) and in maintaining the result at short-term follow-up (at T2). Specifically, the QMR intensive treatment proved to be more effective than the extensive one in terms of “*speed of response*” to the treatment, because mean differences between the sham and intensive groups’ marginal means were always significant in the post hoc analysis. Another important aspect to consider is that no undesirable effects were found during the physiotherapy treatment, sometimes associated with this type of instrumental physical technology, such as an increase in heat in the treated area leading to the interruption of the treatment, burns, or other side effects, with a good adherence to the treatment (dropout < 20%).

The understanding of the interaction between physical agents and biological systems is particularly complex and depends on waveform, frequency, duration, and energy, on the identification of the dose–response effects, and on the characteristics of the targeted cell/tissue types. The identification of the effects of physical agents in terms of how these can modulate a particular cellular function constitutes the basis for its possible clinical application.

A similar frequency range was used in the literature in musculoskeletal pain, albeit with different biophysical sources. In the study by S. Masiero et al. (2020), all patients (diagnosis of osteoarthritis, neck/back pain, or tendinopathies) underwent ten sessions of percutaneous short-wave diathermy, three times/week, with each session lasting 15–20 min, and using frequencies of 4 or 8 MHz and a heat intensity between 40 and 60 W. The authors suggest that the data reported a positive trend in reducing musculoskeletal pain in the short term [24]. Also, ultrasound waves of 1 MHz frequency were applied for 5 min to the target knee, as was direct current for 10 min for 10 treatment sessions together with 0.4% Dexamethasone sodium phosphate (DEX-P) phonophoresis (PH), with 0.4% DEX-P iontophoresis (ION) therapy compared: both therapeutic modalities were found to be effective and generally well tolerated after 10 treatment sessions [18,25]. In low-grade KOA patients according to the Kellgren–Lawrence classification, six sessions of pulsed electromagnetic field therapy (PEMFT) resulting in significantly better results than low-level laser therapy (LLLT) and an immediate positive effect on pain and physical function [26]. Also, analgesia can be produced by low (4 Hz) and high frequency (100 Hz) transcutaneous electrical nerve stimulation (TENS) that is mediated by the release of mu- or delta-opioids, respectively, in the central nervous system [27]. On the other hand, the application of 4.4 MHz of pulsed radiofrequency inhibited pain-related behavior and decreased inflammatory cytokine expression in the inflamed knee joints of rats [28].

Current evidence suggests that low-frequency (≤100 Hz) pulsed subsensory threshold electrical stimulation produced either through PEMF/PES or sham PEMF/PES is effective in improving physical function but not pain intensity at treatment completion in KOA [29].

Again, the Pulsed Electromagnetic Field (PEMF) stimulated the expression of Aggrecan while inhibiting the expression of IL-1β, ADAMTS4, and MMP13. This effect could postpone cartilage degeneration and maintain the microarchitecture of the subchondral trabecular bone in mice. Concurrently, QMR notably decreased NLRP3 levels and activated caspase-1 protein expression, leading to a downregulation of IL-18 and IL-1β protein expression and secretion. Moreover, our research suggests that QMR treatment prompts a shift in macrophage polarization from the M1 to the M2 phenotype in vitro [16,30].

IL-6 and TNFα, as commonly assessed cytokines, appear to be associated with KOA pain. However, cytokine levels may be influenced by various factors, such as age, gender, BMI, disease stage, and synovial inflammation. For instance, Perruccio et al. and Azim et al. identified a positive correlation between serum IL-6 levels and disability pain scores in females but not in males [31,32]. Nevertheless, the experience of pain is intricate and affected by a multitude of factors, including genetics, coping mechanisms, catastrophizing, psychosocial aspects, and sensitization, all of which can influence associations [33,34]. Different inflammatory markers are linked to knee pain, yet correlation data range from weak to moderate, and the quality of evidence varies from conflicting to moderate [30,35].

Our results are in line with the literature and are promising because, with a short protocol, with good patient compliance, at three sessions per week for a total of six sessions, each lasting 30 min, we observed not only a positive result in terms of statistical significance for the VAS scale but, above all, in terms of the severity of pain reported, “mild pain” was 11 mm (95%CI 4 mm to 18 mm), “moderate pain” was 14 mm (95% CI 10 mm to 18 mm), and severe pain was 10 mm (95% CI 6 mm to 14 mm) (see Table 7) [36]. Helping to quickly sham pain, without resorting to drugs, allows patients to be quickly directed to rehabilitative exercise, making QMR a facilitator for the subsequent rehabilitation process. Exercise is recommended in KOA, and patient initiation of and adherence to exercise is key to the success of managing symptoms, whereas many facilitators are related to reinforcement, such as pain reduction, kinesiophobia, sedentary lifestyle, overweight, and other factors [37].

Pain catastrophizing predicts higher pain levels, whereas fear of movement predicts poorer functioning because multiple psychological factors are associated with the development of disability and longer-term worse pain in KOA [38].

In fact, exercise, education, and weight-management are strongly recommended in KOA as clinical practice guidelines [39]: the highly recommended rehabilitation opinions include aerobic exercise programs, weight sham, self-education and management, gait/walking aids, and tai chi. However, the orthopedic insole and hot/cold therapy roles remain controversial [40].

Prolonged applications of continuous ultrasound combined with exercises are effective in providing pain, mobility, functionality, and activity in subjects with knee osteoarthritis [41].

Another important piece of data from our research is the improvement in function in parallel with the reduction of pain, which was not only due to a direct effect, which led the patient to move better and free from pain, but also to the reduction of inflammation, which helped to improve function. The improvement in function in KOA is not always observed in the literature when using instrumental physical therapy: for example, a recent study, involving 3-week treatment of Transcutaneous Electric Nerve Stimulation (TENS) in KOA, found that pain was not reduced compared to the placebo [40]; also, TENS was not effective for stiffness in KOA [42]. Also, high-intensity laser therapy (HILT) was found to be more effective on the stiffness subscale of the Western Ontario and McMaster Universities Osteoarthritis (WOMAC), and combined with exercise therapy, as a useful therapeutic approach, could have positive influences on KOA patients [43].

### Strengths and Weaknesses

This research represents the first patient model to test QMR technology, as a non-ionizing, low-power technology that uses high-frequency waves in the range between 4 and 64 MHz in the rehabilitative treatment of KOA, with promising results for future development ideas. Certainly, further investigation of the effect in other musculoskeletal pain is required, expanding the sample size and investigating the response to treatment in light of protocols combined with rehabilitative physical exercise.

Furthermore, during the treatment we recorded good compliance, with only three dropouts for personal and work reasons, and no side/adverse effects related to the treatment with QMR were recorded.

On the other hand, we need to consider some limitations, as we did not summarize the patients’ medications prior to performing the interventions and did not limit their use during the intervention period; this may have potentially affected the results. Furthermore, since the study included only individuals with grades 1, 2, and 3 KOA, the results of the current study cannot be generalized to individuals whose KOA > 4. Finally, long-term follow-up was not performed; thus, the long-term effects of the interventions could not be determined.

## 5. Conclusions

The results suggest that rehabilitative treatment by QMR in KOA might be an efficacious novel treatment to reduce pain and improve function of patients. The use of QMR as instrumental physical therapy would first of all allow a more rapid improvement of symptoms and an earlier start on the path to rehabilitation exercise and would guarantee greater patient compliance with the rehabilitation plan.

The precise understanding of inflammation’s role in KOA remains unclear, and the question remains whether the inflammatory response causes the progression in osteoarthritis (OA) or the response itself is the consequence [44]. Our previous results suggest an anti-inflammatory effect of QMR technology in a in vitro model of osteoarthritis-related inflammation [45].

Future studies will be necessary to investigate further treatment algorithms and therapeutic associations with therapeutic exercise in an integrated long-term rehabilitation plan.

## Figures and Tables

**Figure 1 diagnostics-14-01200-f001:**
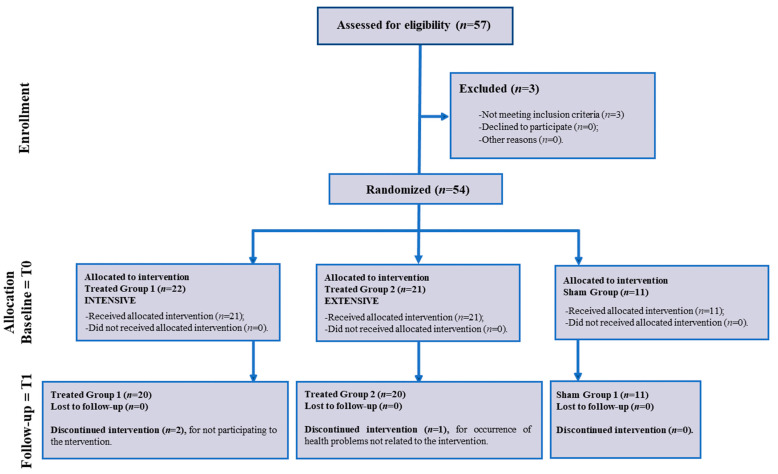
Flow chart of the study.

**Figure 2 diagnostics-14-01200-f002:**
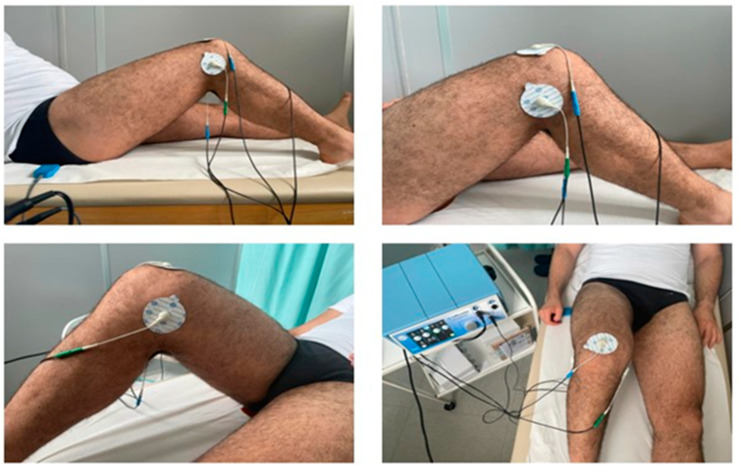
**Arrangement of electrodes.** Electrode No. 1: on the skin region corresponding to the supra-patellar area; electrode No. 2: on the skin region corresponding to the area between the medial femoral condyle and the medial tibial condyle; electrode No. 3: on the skin region corresponding to the area between the lateral femoral condyle and the lateral tibial condyle; electrode No. 4: on the skin region corresponding to the popliteal area.

**Figure 3 diagnostics-14-01200-f003:**
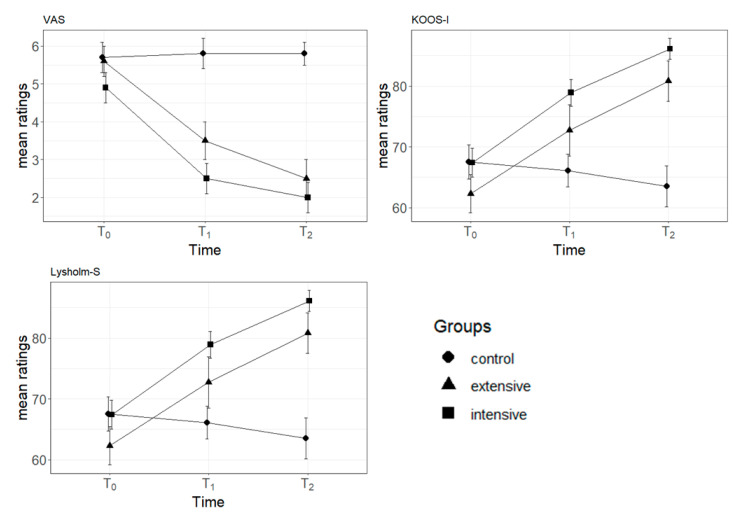
Mean scores of VAS, KOOS-I, and Lysholm-S in relation to time periods. Separate lines indicate different groups. Error bars indicate standard errors. *Note:* VAS = Visual Analog Scale; KOOS-I = Knee Osteoarthritis Outcome Score-Injury: Lysholm-S = Lysholm Knee Scoring Scale. T0 = pre-treatment period; T1 = post-treatment period; T2 = follow-up period.

**Table 1 diagnostics-14-01200-t001:** Descriptive statistics (mean, SD, skewness, and kurtosis) of VAS score, KOOS total score, and Lysholm Scale total score in relation to the three different groups (sham, extensive, and intensive) and time periods (T0, T1, and T2) (See also Figure 2 and Table 2).

Group	Time Periods	Scales	Mean	SD	Skewness	Kurtosis
**Sham**	T_0_	VAS	5.727	1.170	0.471	−0.084
		KOOS-I	67.545	9.136	0.288	−0.517
		Lysholm-S	61.545	16.207	0.542	−0.961
	T_1_	VAS	5.764	1.218	0.504	−0.708
		KOOS-I	66.091	9.115	0.270	0.355
		Lysholm-S	59.455	18.376	0.320	−0.845
	T_2_	VAS	5.791	1.145	−0.148	1.531
		KOOS-I	63.545	11.273	−0.128	−1.061
		Lysholm-S	59.364	16.200	0.408	−0.398
**Extensive**	T_0_	VAS	5.600	1.650	0.360	−0.156
		KOOS-I	62.350	13.963	−0.068	−0.306
		Lysholm-S	53.900	20.460	−0.216	−0.869
	T_1_	VAS	3.495	2.196	1.086	1.259
		KOOS-I	72.650	18.639	−1.723	4.137
		Lysholm-S	72.050	20.062	−0.803	0.624
	T_2_	VAS	2.545	2.057	0.693	−0.402
		KOOS-I	80.800	14.742	−1.770	4.332
		Lysholm-S	80.550	18.650	−1.324	1.063
**Intensive**	T_0_	VAS	4.945	1.596	0.480	−0.870
		KOOS-I	67.350	10.599	−0.586	−0.248
		Lysholm-S	65.000	14.480	−0.637	−0.376
	T_1_	VAS	2.515	1.713	0.171	−1.178
		KOOS-I	78.950	9.736	−0.745	0.263
		Lysholm-S	82.300	11.193	−0.084	−1.425
	T_2_	VAS	1.975	1.626	0.804	−0.328
		KOOS-I	86.050	7.715	−0.131	−0.062
		Lysholm-S	88.250	10.078	−1.045	0.886

*Note:* VAS = Visual Analog Scale; KOOS-I = Knee Osteoarthritis Outcome Score-Injury: Lysholm-S = Lysholm Knee Scoring Scale. T_0_ = pre-treatment period; T_1_ = post-treatment period; T_2_ = follow-up period. SD = standard deviation.

**Table 2 diagnostics-14-01200-t002:** Descriptive statistics (mean, SD, skewness, and kurtosis) for individual characteristics (age, years from diagnosis, physical activity in hours per week, and BMI).

Individual Characteristics	Mean	SD	Skewness	Kurtosis
Age	64.61	11.09	−0.23	−0.48
years from diagnosis	61.57	10.98	−0.26	−0.74
physical activity	2.30	3.74	3.06	11.43
BMI	26.91	6.15	1.14	1.33

*Note*: BMI = Body Mass Index. SD = standard deviation.

**Table 3 diagnostics-14-01200-t003:** Frequencies and relative percentages of gender, treatment group, working status, civil status, and presence of comorbidity.

Gender	Freq.	Percentage (%)
Male	18	35.3
Female	33	64.7
**Group**	**Freq.**	**Percentage (%)**
Sham	11	21.6
Extensive	20	39.2
Intensive	20	39.2
**working status**	**Freq.**	**Percentage (%)**
Unemployed	32	62.7
Employed	19	37.3
**civil status**	**Freq.**	**Percentage (%)**
Married	41	80.4
Single	10	19.6
**Comorbidity**	**Freq.**	**Percentage (%)**
No	20	39.2
Yes	31	60.8

**Table 4 diagnostics-14-01200-t004:** ANOVA table of 3 (time series) × 3 (treatment of gonarthrosis) mixed ANOVA design for VAS scores. η^2^ and ω^2^ are effect sizes.

Within-Subject Effects							
Cases	Sum of Squares	df	Mean Square	F	*p*(F)	η^2^	ω^2^
Time series	101.122	2.000	50.561	45.061	<0.001	0.142	0.187
Time series × treatment of gonarthrosis	45.129	4.000	11.282	10.055	<0.001	0.063	0.070
Residuals	107.718	96.000	1.122				
**Between-Subject Effects**							
**Cases**	**Sum of Squares**	**df**	**Mean Square**	**F**	***p*(F)**	**η^2^**	**ω^2^**
treatment of gonarthrosis	147.016	2.000	73.508	11.306	<0.001	0.206	0.123
Residuals	312.090	48.000	6.502				

*Note:* Type III Sum of Squares.

**Table 5 diagnostics-14-01200-t005:** ANOVA table of 3 (time series) × 3 (treatment of gonarthrosis) mixed ANOVA design for KOOS-I scores. η^2^ and ω^2^ are effect sizes.

Within-Subject Effects							
Cases	Sum of Squares	df	Mean Square	F	*p*(F)	η^2^	ω^2^
Time series	2930.443	2.000	1465.221	32.973	<0.001	0.096	0.110
Time series × treatment of gonarthrosis	2215.100	4.000	553.775	12.462	< 0.001	0.072	0.070
Residuals	4265.985	96.000	44.437				
**Between-Subject Effects**							
**Cases**	**Sum of Squares**	**df**	**Mean Square**	**F**	***p*(F)**	**η^2^**	**ω^2^**
Treatment of gonarthrosis	2992.923	2.000	1496.462	3.953	0.026	0.098	0.039
Residuals	18,171.129	48.000	378.565				

*Note:* Type III Sum of Squares.

**Table 6 diagnostics-14-01200-t006:** ANOVA table of 3 (time series) × 3 (treatment of gonarthrosis) mixed ANOVA design for Lysholm-S scores. η^2^ and ω^2^ are effect sizes.

Within-Subject Effects							
Cases	Sum of Squares	df	Mean Square	F	*p*	η^2^	ω^2^
Time series	6278.795	2.000	3139.398	46.763	<0.001	0.111	0.133
Time series × treatment of gonarthrosis	3462.777	4.000	865.694	12.895	<0.001	0.061	0.064
Residuals	6444.818	96.000	67.134				
**Between-Subject Effects**							
**Cases**	**Sum of Squares**	**df**	**Mean Square**	**F**	** *p* **	**η^2^**	**ω^2^**
Treatment of gonarthrosis	7567.730	2.000	3783.865	5.557	0.007	0.134	0.058
Residuals	32,686.832	48.000	680.976				

*Note:* Type III Sum of Squares.

**Table 7 diagnostics-14-01200-t007:** Post hoc analysis (Tukey test) between marginal means of treatment groups for VAS. KOOS-I and Lysholm-S for T0, T1, and T2.

Time Series	Scales	Group Comparison	Mean Difference	SE	t	Cohen’s d	p_tukey_	
	VAS	sham vs. extensive	0.127	0.641	0.199	0.075	1.000	
		sham vs. intensive	0.782	0.641	1.221	0.450	0.950	
		extensive vs. intensive	0.655	0.540	1.213	0.384	0.951	
T0	KOOS-I	sham vs. extensive	5.195	4.686	1.109	0.416	0.971	
		sham vs. intensive	0.195	4.686	0.042	0.016	1.000	
		extensive vs. intensive	−5.000	3.947	−1.267	−0.401	0.938	
	Lysholm-S	sham vs. extensive	7.645	6.188	1.236	0.464	0.945	
		sham vs. intensive	−3.455	6.188	−0.558	−0.210	1.000	
		extensive vs. intensive	−11.100	5.213	−2.129	−0.673	0.463	
	VAS	sham vs. extensive	2.269	0.641	3.540	1.329	0.018	**
		sham vs. intensive	3.249	0.641	5.096	1.903	<0.001	***
		extensive vs. intensive	0.980	0.540	1.815	0.574	0.672	
T1	KOOS-I	sham vs. extensive	−6.559	4.686	−1.400	−0.525	0.894	
		sham vs. intensive	−12.859	4.686	−2.744	−1.030	0.151	
		extensive vs. intensive	−6.300	3.947	−1.596	−0.505	0.804	
	Lysholm-S	sham vs. extensive	−12.595	6.188	−2.035	−0.764	0.525	
		sham vs. intensive	−22.845	6.188	−3.692	−1.386	0.012	**
		extensive vs. intensive	−10.250	5.213	−1.966	−0.622	0.572	
	VAS	sham vs. extensive	3.246	0.641	5.064	1.901	<0.001	***
		sham vs. intensive	3.816	0.641	5.954	2.235	<0.001	***
		extensive vs. intensive	0.570	0.541	1.056	0.334	0.979	
T2	KOOS-I	sham vs. extensive	−17.255	4.686	−3.682	−1.382	0.013	**
		sham vs. intensive	−22.505	4.686	−4.803	−1.803	<0.001	***
		extensive vs. intensive	−5.250	3.947	−1.330	−0.421	0.919	
	Lysholm-S	sham vs. extensive	−21.186	6.188	−3.424	−1.285	0.027	**
		sham vs. intensive	−28.886	6.188	−4.668	−1.752	<0.001	***
		extensive vs. intensive	−7.700	5.213	−1.477	−0.467	0.862	

*Note:* ** *p* < 0.01, *** *p* < 0.001. Results are averaged over the levels of time periods T0 (pre-treatment), T1 (post-treatment), and T2 (follow-up). VAS = Visual Analog Scale; KOOS-I = Knee Osteoarthritis Outcome Score-Injury: Lysholm-S = Lysholm Knee Scoring Scale.

**Table 8 diagnostics-14-01200-t008:** Contrast analysis of marginal means of VAS, KOO-I, and Lysholm-S.

Scales	Contrasts	Estimate	SE	df	t	*p*(t)	Effect Size (Cohen’s d)	Effect Size Classes
VAS	linear	−1.405	0.154	96	−9.108	<0.001	−2.551	large
	quadratic	0.413	0.154	96	2.678	0.009	0.750	medium
KOOS-I	linear	7.814	0.971	96	8.048	<0.001	2.254	large
	quadratic	−1.053	0.971	96	−1.085	0.281		
Lysholm-S	linear	11.247	1.193	96	9.425	<0.001	2.640	large
	quadratic	−2.586	1.193	96	−2.167	0.033	−0.607	medium

*Note:* VAS = Visual Analog Scale; KOOS-I = Knee Osteoarthritis Outcome Score-Injury: Lysholm-S = Lysholm Knee Scoring Scale.

## Data Availability

All data are available upon request to the corresponding author.

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
