# Peer review of "Management and Rehabilitative Treatment in Osteoarthritis with a Novel Physical Therapy Approach: A Randomized Control Study"

_diagnostics, 2024, doi:10.3390/diagnostics14111200_

Round 1

Reviewer 1 Report

Comments and Suggestions for Authors

First of all, I would like to thank the editorial team for considering me to review this article. I would like to congratulate the authors for their work. However, there are some aspects that need to be modified before final publication.

ABSTRACT

Even if it is a structured abstract, include everything in a single paragraph.

The second aim is not well defined, it is too general.

The results could be a little more extensive

INTRODUCTION

At the beginning of the introduction (lines 50-51), I would try to link this first paragraph with the following paragraph so that it does not stand alone as an isolated sentence. The same in lines 71-73, 91-93, 94-97.

More articles need to be included in the introduction. For example, lines 52 to 61 are justified by a single article.

What is EULAR and ACR? Please define it.

If previous research has already shown that QMR treatment is effective, what is new in your study? You need to better define the novelty of your research.

I recommend splitting the aims with appendices (a, b, c). Right now it is difficult for the reader to know how many aims are pursued in the research.

Research hypotheses are missing.

MATERIALS AND METHODS

The font of "NCT06239805" seems different from the rest of the manuscript.

Include sample size calculation

At the beginning of the statistical analysis: “Analysis We calculated descriptive statistics”. Modify this.

RESULTS

Just as you include subheading 3.2.1 for power analysis, I think you could divide this results section into more subheadings. There are 5 tables included in this section and it can be difficult to follow.

DISCUSSION

Tables are not often mentioned in the discussion. See table 7" should not be indicated.

The information provided in the discussion is correct. However, it should be based on more previous scientific evidence and compared to a greater extent with that found in the present study.

The limitations and strengths are in line with the article presented.

OVERALL

Some paragraphs are not indented, others are. Please unify.

Further justification of previous research is needed, both in the introduction and in the discussion.

Some sections of the manuscript present the references in Vancouver and others in APA. Unify.

Author Response

First of all, I would like to thank the editorial team for considering me to review this article. I would like to congratulate the authors for their work. However, there are some aspects that need to be modified before final publication.

Comment: ABSTRACT

Even if it is a structured abstract, include everything in a single paragraph.

The second aim is not well defined, it is too general.

The results could be a little more extensive

Response: Dear reviewer, we have modified the abstract following your requests, also considering the words limit required for the formulation of the abstract.

Comment: INTRODUCTION

At the beginning of the introduction (lines 50-51), I would try to link this first paragraph with the following paragraph so that it does not stand alone as an isolated sentence. The same in lines 71-73, 91-93, 94-97.

Response: Thank you, we have done.

Comment: More articles need to be included in the introduction. For example, lines 52 to 61 are justified by a single article.

Response: Dear Reviewer, thank you for the comment. We have added.

Comment: What is EULAR and ACR? Please define it.

Response: We have defined. Thank you.

Comment: If previous research has already shown that QMR treatment is effective, what is new in your study? You need to better define the novelty of your research.

Response: Dear Reviewer, previous researches were related to in vitro anti-inflammatory property. We have better defined in the text.

Comment: I recommend splitting the aims with appendices (a, b, c). Right now it is difficult for the reader to know how many aims are pursued in the research.

Response: We have done. Thank you.

Comment: Research hypotheses are missing.

Response: Thank you. We have added.

MATERIALS AND METHODS

Comment: The font of "NCT06239805" seems different from the rest of the manuscript.

Response: Thank you. We have corrected.

Comment: Include sample size calculation

Response: Thank you. In accordance we have included a subsection on Sample Size calculation.

Comment: At the beginning of the statistical analysis: “Analysis We calculated descriptive statistics”. Modify this.

Response: Dear Reviewer, we have corrected.

RESULTS

Comment: Just as you include subheading 3.2.1 for power analysis, I think you could divide this results section into more subheadings. There are 5 tables included in this section and it can be difficult to follow.

Response: As suggested by the reviewer, we inserted further subheadings in subsection 3.2

DISCUSSION

Comment: Tables are not often mentioned in the discussion. See table 7" should not be indicated.

Response: We have added. Thank you for the comment.

Comment: The information provided in the discussion is correct. However, it should be based on more previous scientific evidence and compared to a greater extent with that found in the present study.

Response: Thanking you, we have enriched the discussion in accordance.

Comment: The limitations and strengths are in line with the article presented.

Response: Thank you

OVERALL

Some paragraphs are not indented, others are. Please unify.

Response: Dear Reviewer, we have corrected.

Further justification of previous research is needed, both in the introduction and in the discussion.

Response: We have added. Thank you.

Some sections of the manuscript present the references in Vancouver and others in APA. Unify.

Response: We have corrected. Thank you.

Reviewer 2 Report

Comments and Suggestions for Authors

In the manuscript “Management and Rehabilitative Treatment in Osteoarthritis with a Novel Physical Therapy Approach: a Randomized Control Study”, the authors investigated the effects of Quantic Magnetic Resonance (QMR) on pain reduction and functional improvement in Knee Osteoarthritis (KOA) patients. using a double-blind, randomized, controlled setup. The results showed that the QMR treatment groups showed a faster response to treatment, with significant improvements in pain and function compared to the sham group, suggesting that QMR could be an effective treatment for slowing KOA progression and improving patient outcomes. The manuscript was well-written overall, and the experimental design is straightforward. My primary comments and suggestions relate to representation and interpretation of the results. These specific comments and suggestions are included below.

Major comments:

  1. It would be better to unify the 'Control' or 'Sham' group throughout the manuscript. Thus, audience could know which group the authors referred to.

  2. The Sham group only has 11 patients. The number is only about half of the other two groups. How did the authors decide the number of patients in each group. On Line 112, the groups were generated 'according to a computer-generated simple randomization list at a 1:1:1 ratio'. The number of patients in the three groups is not in the ratio of 1:1:1. It would be better if the authors could provide more information about his randomization process for clarifications.

  3. Between T1 and T2, did patients take any other rehabilitation procedures or medicines?

  4. The authors aimed to investigate 'the improvement in function' of patients in the Introduction and Discussion. However, it is not very clear how the function was measured and what kind of functions was investigated. I believe the function was measured using the 3 scales. It would be beneficial for the authors to provide more details about the 3 scales on how the pain and function were measured, since the all the results are from these scales.

  5. The statistical analysis was performed on the results from the 3 scales. For table 7, why did the authors average the results over the 3 time periods and perform the comparisons? I think it might be better to do pairwise comparisons with corrections to show if there were any differences among 3 groups at T0 (make sure the baseline scores are not significantly different before treatment) and then investigate the differences among the 3 groups at T1 and T2 to show the effect of the treatment. In addition, from Figure 3, at T0, we could see the difference in scores from the 3 scales between the extensive and intensive groups before the treatment. It is not fair to say one treatment is more efficient than another if the patients have significant different scores before the treatments (Line 280 'Table 7 shows that the intensive treatment is the most efficient, because mean differences between control and intensive groups marginal means are always significant in the hoc analysis.'). Thus, I think that showing the statistics at T0 is important to ensure patients in the 3 groups have similar scores before treatments and it might be better to investigate the relative changes in the scores (instead of the absolute scores) to show which treatment is more efficient. From Figure3, the two treatments seem to have similar effects.

  6. In the discussion, the authors talked about frequency ranges used in different therapies and 'our findings indicate that QMR treatment induces a switch in macrophage polarization from the M1 phenotype to the M2 phenotype in vitro'. I think the results could only show QMR might be a treatment to reduce paint and improve function of patients. It is unclear how the results could lead to the conclusions the authors made in the discussions. It would be great for the authors to provide more evidence to support their conclusions and be careful about the conclusions they could draw from the results.

Minor comments:

Line 71: Definitions of EULAR and ACR were missing.

Line 98, What are the assumptions here?

Line 193: 'scale14' or 'scale [14]'.

Line 194: No space between 'con' and 'sists'

Line 199: 'scale15' or 'scale [15]'.

Line 208. References for the literature.

Line 213: Definitions of T0-T3 were unnecessary since they were already well defined in the Methods.

Line 355. Table 7 did not provide info about the MCSD.

Comments on the Quality of English Language

Minor editing of English language is required.

The authors used a lot of ';' incorrectly throughout the manuscript. 

Author Response

In the manuscript “Management and Rehabilitative Treatment in Osteoarthritis with a Novel Physical Therapy Approach: a Randomized Control Study”, the authors investigated the effects of Quantic Magnetic Resonance (QMR) on pain reduction and functional improvement in Knee Osteoarthritis (KOA) patients. using a double-blind, randomized, controlled setup. The results showed that the QMR treatment groups showed a faster response to treatment, with significant improvements in pain and function compared to the sham group, suggesting that QMR could be an effective treatment for slowing KOA progression and improving patient outcomes. The manuscript was well-written overall, and the experimental design is straightforward. My primary comments and suggestions relate to representation and interpretation of the results. These specific comments and suggestions are included below.

Major comments:

It would be better to unify the 'Control' or 'Sham' group throughout the manuscript. Thus, audience could know which group the authors referred to.

Response: We have chosen “Sham” and corrected in accordance. Thank you.

The Sham group only has 11 patients. The number is only about half of the other two groups. How did the authors decide the number of patients in each group. On Line 112, the groups were generated 'according to a computer-generated simple randomization list at a 1:1:1 ratio'. The number of patients in the three groups is not in the ratio of 1:1:1. It would be better if the authors could provide more information about his randomization process for clarifications.

Response: The control group has less participants to allow more individuals to benefit from rehabilitation (in the control group no outpatients did not receive any benefit). Therefore, we paid more attention to created balanced experimental groups to compare them with a smaller control group. Higher control groups not necessarily guarantee higher power in statistical test, especially if effects sizes are small or high. For F test used in ANOVA, in particular, if samples have similar variances even if different sizes, an increment of the size of control group does not improve the statistical power (see: Oldfield, M. (2016). Unequal sample sizes and the use of larger control groups pertaining to power of a study. Ministry of Defence UK Paper : DSTLTR92592 P2PP2R-2016-02-23T13. DOI: 10.6084/m9.figshare.23988477.). We performed different Levene’s tests for homogeneity of variance between control and experimental groups and they resulted non significant. Therefore we set the size of the control group half size in relation to experimental groups.

Between T1 and T2, did patients take any other rehabilitation procedures or medicines?

Response: They didn’t. Thank you.

The authors aimed to investigate 'the improvement in function' of patients in the Introduction and Discussion. However, it is not very clear how the function was measured and what kind of functions was investigated. I believe the function was measured using the 3 scales. It would be beneficial for the authors to provide more details about the 3 scales on how the pain and function were measured, since the all the results are from these scales.

Response: Thanks for your comment. We specified already in the introduction and in the discussion that the recovery of function was under-stood as "in activities of daily living" and walking for the KOOS and Lysholm scales.

2.4 Outcome measures

The following scales were used to assess pain and function at T0, T1 and T2: Visual Analogue Scale (VAS) [13]; Knee Injury and Osteoarthritis Outcome Scale (KOOS) [14]; Lysholm knee scoring scale [15].

Pain evaluation:

The VAS scale is the most widely used tool for pain assessment [13] and consists of a one-dimensional rating of pain intensity; it is a continuous scale consisting of a horizontal or vertical line, generally 10 cm (100 mm) long, with two start and end points marked 'no

pain' and 'worst pain ever'.

Functional evaluation:

Recovery of function was understood as improvement in symptoms, walking and improvement in common activities of daily living and quality of life for the Knee injury and Osteoarthritis Outcome Score (KOOS-I) [14] and Lysholm Knee Scoring (Lysholm-S) [15].

The statistical analysis was performed on the results from the 3 scales. For table 7, why did the authors average the results over the 3 time periods and perform the comparisons? I think it might be better to do pairwise comparisons with corrections to show if there were any differences among 3 groups at T0 (make sure the baseline scores are not significantly different before treatment) and then investigate the differences among the 3 groups at T1 and T2 to show the effect of the treatment. In addition, from Figure 3, at T0, we could see the difference in scores from the 3 scales between the extensive and intensive groups before the treatment. It is not fair to say one treatment is more efficient than another if the patients have significant different scores before the treatments (Line 280 'Table 7 shows that the intensive treatment is the most efficient, because mean differences between control and intensive groups marginal means are always significant in the hoc analysis.'). Thus, I think that showing the statistics at T0 is important to ensure patients in the 3 groups have similar scores before treatments and it might be better to investigate the relative changes in the scores (instead of the absolute scores) to show which treatment is more efficient. From Figure3, the two treatments seem to have similar effects.

Response: we changed table 7 for post-hoc analyses differentiating group comparisons for each time point T0, T1 and T2. We also changed the paragraph: “Table 7 shows that the intensive treatment is the most efficient in the post-hoc analysis, because mean differences between control and intensive groups marginal means are significant in T1 and T2 and have larger effect sizes (Cohen’s d) in comparison to the differences between control and extensive groups. Before the treatment (T0) post-hoc analysis revealed no differences between groups.”

In the discussion, the authors talked about frequency ranges used in different therapies and 'our findings indicate that QMR treatment induces a switch in macrophage polarization from the M1 phenotype to the M2 phenotype in vitro'. I think the results could only show QMR might be a treatment to reduce paint and improve function of patients. It is unclear how the results could lead to the conclusions the authors made in the discussions. It would be great for the authors to provide more evidence to support their conclusions and be careful about the conclusions they could draw from the results.

Response: Thanks for your comment. We have reshaped the conclusion and the assumptions in the conclusions according to your indications. Specifically, the action of QMR in vitro as an anti-inflammatory in reference to macrophage M1 - M2 has been previously published. For the future, the objective will be to evaluate the anti-inflammatory and regenerative action on synovial tissue.

Paolucci, T.; Pino, V.; Elsallabi, O.; Gallorini, M.; Pozzato, G.; Pozzato, A.; Lanuti, P.; Reis, V.M.; Pesce, M.; Pantalone, A.; et al. Quantum Molecular Resonance Inhibits NLRP3 Inflammasome/Nitrosative Stress and Promotes M1 to M2 Macrophage Polarization: Potential Therapeutic Effect in Osteoarthritis Model In Vitro. Antioxidants 2023, 12: 1358. https://doi.org/10.3390/antiox12071358

Minor comments:

Line 71: Definitions of EULAR and ACR were missing. We have included. Thank you.

Line 98, What are the assumptions here?

Line 193: 'scale14' or 'scale [14]'. Is [14], we have corrected.

Line 194: No space between 'con' and 'sists'. We have corrected. Thank you.

Line 199: 'scale15' or 'scale [15]'. Is [15], thank you.

Line 208. References for the literature. Thank you.

Line 213: Definitions of T0-T3 were unnecessary since they were already well defined in the Methods. We have eliminated. Thank you.

Line 355. Table 7 did not provide info about the MCSD. Thank you. We have corrected the discussion.

Comments on the Quality of English Language

Minor editing of English language is required.

The authors used a lot of ';' incorrectly throughout the manuscript. 

The manuscript has been revised by a english mother tongue expert in scientific language. Thank you

Round 2

Reviewer 1 Report

Comments and Suggestions for Authors

Although the introduction has been expanded in line with the comments, the discussion still lacks justification. The modifications made by the authors have been rather superficial.

As for paragraph indentation, there are still some paragraphs that are indented and others that are not.

Author Response

Reviewer: Although the introduction has been expanded in line with the comments, the discussion still lacks justification. The modifications made by the authors have been rather superficial.

Response: Thank you for the comment. We have revised the discussion in accordance.

Reviewer: As for paragraph indentation, there are still some paragraphs that are indented and others that are not.

Response: It is true. Thanking you, we have corrected it.